

# Nature-derived lignan compound VB-1 exerts hair growth-promoting effects by augmenting Wnt/β-catenin signaling in human dermal papilla cells

Jieshu Luo[1,2], Mengting Chen[1,2], Yingzi Liu[1,2], Hongfu Xie[1], Jian Yuan[3], Yingjun Zhou[4], Jinsong Ding[4], Zhili Deng[1,2,5] and Ji Li[1,2,5]

[1] Department of Dermatology, Xiangya Hospital, Central South University, Changsha, China
[2] Center for Molecular Medicine, Xiangya Hospital, Central South University, Changsha, China
[3] Department of Neurosurgery, Xiangya Hospital, Central South University, Changsha, China
[4] Department of Pharmacology, School of Pharmaceutical Sciences, Central South University, Changsha, China
[5] Key Laboratory of Organ Injury, Aging and Regenerative Medicine of Hunan Province, Central South University, Changsha, China

Corresponding authors
Zhili Deng, dengzhili@csu.edu.cn
Ji Li, liji0704@163.com,
liji_xy@csu.edu.cn

## ABSTRACT

**Background**. Vitexin is a kind of lignan compound which has been shown to possess a variety of pharmacological effects, such as anti-inflammatory, anti-oxidative and anti-cancer activities. However the effect of vitexin on hair regeneration has not been elaborated.

**Methods**. The proliferation of human dermal papilla cells (hDPCs) was examined by cell counting and continuous cell culture after vitexin compound 1 (VB-1) was treated. The expression of *lef1*, *wnt5a*, *bmp2*, *bmp4*, *alpl* and *vcan* was examined by RT-PCR. The expression of *dkk1*, t*gf-β1*, active-β-Catenin, and AXIN2 was examined by RT-PCR or immunoblotting. Hair shaft growth was measured in the absence or presence of VB-1.

**Results**. We demonstrated that VB-1 significantly promotes the proliferation of hDPCs in a concentration-dependent manner within a certain concentration range. Among the hair growth-related genes investigated, *dkk1* was clearly down-regulated in hDPCs treated with VB-1. The increased active β-Catenin and decreased AXIN2 protein levels suggest that VB-1 facilitates Wnt/β-catenin signaling in hDPCs *in vitro*. The expression of DP signature genes was also upregulated after VB-1 treatment. Our study further indicated that VB-1 promotes human hair follicle (HF) growth by HF organ culture assay.

**Discussion**. VB-1 may exert hair growth-promoting effects via augmenting Wnt/β-catenin signaling in hDPCs.

# INTRODUCTION

The hair follicle (HF) is a complex mini-organ composed of epidermal and mesenchymal (dermal) components which undergo cycles of degeneration (catagen), rest (telogen), and growth (anagen) throughout adult life (*Greco et al., 2009*). This hair cycle is based on the capacity of hair follicle stem cells (HFSCs), which are slow-cycling, label-retaining cells

located at a niche known as the bulge, to transiently exit the quiescent status to launch the growth phase (*Kandyba et al., 2013*; *Tumbar et al., 2004*). Activation and differentiation of HFSCs are mainly governed by a cluster of specialized mesenchymal cells residing in the base of hair follicles, known as the dermal papilla (DP) (*Morgan, 2014*). With anagen initiation, stem cells in the bulge are activated to fuel the growth of new hair follicles in response to DP signals (*Kobielak et al., 2007*; *Tang et al., 2016*).

During the postnatal hair cycle, dermal papilla cells (DPCs) act as a signaling center to control the proliferation, migration, and differentiation of the surrounding epithelial stem/progenitor cells to complete the process of hair regeneration. Moreover, DPCs possess hair follicle-inducing ability via interacting with neighboring epithelial stem cells (*Aoi et al., 2012*; *Higgins et al., 2013*; *Ohyama et al., 2012*). Several signaling pathways, particularly Wnt/β-catenin signaling, have been shown to play a key role in the development of new hair follicles and initiation of hair growth (*Kandyba et al., 2013*; *Chu et al., 2004*; *Lim & Nusse, 2013*; *Lei et al., 2017*; *Lei & Chuong, 2016*; *Lei, Yang & Chuong, 2017*). Numerous Wnt ligands and inhibitors expressed in DPCs are crucial for regulating hair growth (*Plikus, 2012*; *Kwack et al., 2012*; *Kwack et al., 2008*; *Lei et al., 2014*; *Lei, Chuong & Widelitz, 2013*; *He et al., 2017*). The proper crosstalk between the mesenchyme and epithelium facilitates the activation of HFSCs by overcoming the repressive signals that maintain HFSCs in a quiescent state (*Morgan, 2014*; *Oshimori & Fuchs, 2012*; *Deng et al., 2015*). Genetic deletion of β-catenin in the DP results in premature induction of catagen and prevents regeneration of HFs (*Enshell-Seijffers et al., 2010*).

Several hair disorders are characterized by the inability to re-enter the regeneration phase (anagen) of the hair cycle. Particularly, in the case of androgenetic alopecia, ectopic activation of androgen receptor signaling responding to dihydrotestosterone in the HF, mainly in the DP, alters the expression of hair growth-related paracrine factors (such as DKK1, Wnts, and TGF-βs). Dysregulation of these paracrine factors impairs the proliferation and differentiation of hair follicle stem/matrix cells, causing shortening of the anagen phase and resulting in progressive HF miniaturization, a major characteristic of androgenetic alopecia (*Kwack et al., 2008*; *Inui & Itami, 2011*; *Shin et al., 2013*; *Ceruti, Leirós & Balañá, 2017*; *Hu et al., 2012*). Therefore, DP is thought to be the primary therapy target for androgenetic alopecia. Current pharmacological treatment for androgenetic alopecia is mainly concentrated on the prevention of further hair loss (*Varothai & Bergfeld, 2014*). However, the development of pharmacologic agents to activate the proliferation of HFSCs and reboot the hair cycle has been unsatisfactory.

Lignan is a group of complex polyphenolic antioxidants widely present in plant vitex negundo (*Adlercreutz, 2007*; *Adlercreutz, 2002*). Vitexin is a kind of lignan compound found in vitex negundo seeds, widely used in herbal medicine in China (*Zhou et al., 2009*; *Xin et al., 2013*). Vitexin has been shown to possess a variety of pharmacological effects, such as anti-inflammatory, anti-oxidative and anti-cancer activities (*Yang et al., 2014*). Clinical studies indicated that lignans have a potential role in cancer prevention (*Wang et al., 2014*; *Thompson et al., 2005*). Some clinical trials have confirmed that lignans can also inhibit the development of certain cancers. For example, some studies have shown that a low risk of ovarian cancer and prostate cancer is correlated with a high lignan

intake diet. This may be the reason why mediterranean diets (olives have high lignan content) are associated with a lower incidence of cancer. The anti-cancer properties of lignans have been studied in cancer cells culture *in vitro*. In these studies, purified vitexin compound-1(VB-1) prevented the proliferation of cancer cells in the G2/M phase of the cell cycle and induced apoptosis within a certain concentration range effectively (*Zhou et al., 2009*; *Xin et al., 2013*). Although VB-1 has been shown to exert anti-cancer activities, other pharmacological effects are still unclear. It is worth further study of their pharmacological values.

Here, we evaluated the effect of VB-1 on hair growth. We demonstrated that VB-1 facilitated the proliferation of hDPCs in a concentration-dependent manner within a certain concentration range. Among the hair growth-related genes investigated, *dkk1* significantly decreased in VB-1-treated hDPCs. By immunoblot analysis, we showed that active β-Catenin increased and AXIN2 decreased, suggesting that VB-1 promotes Wnt/β-catenin signaling in hDPCs. Furthermore, VB-1 enhances the expression of DP signature genes in hDPCs. Moreover we found that VB-1 promoted human HF growth in an organ culture assay. Taken together, these findings indicate that VB-1 promotes hair growth and may be a new therapy for hair loss treatment.

## MATERIALS AND METHODS

### Reagents

Vitexin compound-1(VB-1), was kindly provided by Prof. Yingjun Zhou and prepared as previously described (*Zhou et al., 2009*). VB-1 powder was produced in Prof. Jinsong Ding's lab (Department of Medicinal Chemistry, Central South University, China) and used in this study.

### Isolation and culture of human hair follicles

Punch scalp biopsy (5 mm) specimens were obtained from male non-balding occipital scalps of patients undergoing hair transplantation surgery for androgenetic alopecia. All procedures involving human subjects were approved by the Institutional Review Board of Xiangya hospital (IRB NO. 201611609) in accordance with the Helsinki guidelines. Briefly, hair follicles were isolated with scissors and forceps under a binocular light microscope and cultured in 24-well dishes for 14 days in William's E medium (Gibco, Grand Island, NY, USA) supplemented with 10 mg/mL insulin, 2 mM L-glutamine, 10 ng/mL hydrocortisone, and 100 U/mL streptomycin at 37 °C in a 5% (v/v) $CO_2$ atmosphere (*Fischer, Hipler & Elsner, 2007*). Hair follicles were incubated in William's E medium with VB1, and were photographed by immersing in PBS at 37 °C, using a stereoscope every 48 h. In all experiments, VB-1 culture medium was refreshed every other day. A total of 150 anagen hair follicles were isolated from three different volunteers and cultured with each concentration of VB-1, and the experiments were repeated five times with six repetitions for each concentration group.

### Isolation and culture of hDPCs

The hDPCs were isolated and cultured as previously described (*Gledhill, Gardner & Jahoda, 2013*). Briefly, isolated hDPCs were cultured in Dulbecco's modified Eagle's medium

(Gibco) supplemented with 10% (v/v) fetal bovine serum (Gibco), 100 U/mL penicillin, and 100 mg/mL streptomycin. The medium was changed every 2 days. Once cell outgrowth was sub-confluent, hDPCs were harvested with 0.25% (w/v) trypsin-EDTA (Invitrogen, Carlsbad, CA, USA) and passage cultured with a split radio of 1:3. hDPCs at passages 3–5 were used in this study.

## MTS assay

MTS working solution was added as 20 $\mu$l/well to the culture wells, after cell seeding for 4 h, then shaken and mixed. The 96-well plates were incubated in a 37 °C, 5% $CO_2$ incubator for 1 h. Value of $OD_{490nm}$ was recorded at the subsequent 12 h, 24 h, 48 h, and 72 h time points. The growth curve was plotted by $OD_{490nm}$ value.

## Total RNA isolation, cDNA synthesis, and real-time PCR

RNA was extracted from cells using TRIzol reagent (Invitrogen) and cDNA was prepared using the RevertAid First Strand cDNA Synthesis Kit (Gibco/Thermo Scientific, Waltham, MA, USA). Sequences of qPCR primers were from PrimerBank and queried in NCBI blast to check their specificity. *gapdh* was used as an internal reference. For RT-PCR 1 $\mu$l each of the forward primer (10 ng/ $\mu$l) and Reverse Primer (10 ng/ $\mu$l) per well, iTaq Universal Green Supermix ($2\times$) 10 $\mu$l, Nuclease-Free $H_2O$ 6 $\mu$l, and cDNA 2 $\mu$l were mixed to a final volume of 20 $\mu$l, The PCR procedure software recorded the average fluorescence value of each cycle of the reaction. The relative expression levels of different genes in the cells were obtained by comparing the Ct values. The experiment was repeated three times with three replicates seted per reaction. PCR primer sequences are given in Table S1.

## Western blot assays

The collected cells were lysed in RIPA buffer (Thermo Scientific) with protease inhibitors (Thermo Scientific) after rinsing twice with PBS (precooled at 4 °C). Proteins were quantified in the bicinchoninic acid assay (Pierce BCA Protein Assay, Rockford, IL, USA), and then separated by SDS–polyacrylamide gel electrophoresis and electroblotted onto polyvinylidene fluoride membranes. After blocking in 5% nonfat milk 1 h at 25 °C, the membranes were probed with primary antibodies followed by horseradish peroxidase-conjugated secondary antibodies (Santa Cruz Biotechnology, Dallas, TX, USA). Primary antibodies used were rabbit anti-active β-Catenin protein (1:1,000, Cell Signaling Technology, Danvers, MA, USA), rabbit anti-AXIN2 (1:1,000, Cell Signaling Technology) and mouse anti-Tubulin (1:1,000, Cell Signaling Technology). Immunoreactive bands were visualized with horseradish peroxidase substrate (Luminata, Millipore, Billerica, MA, USA) using the ChemiDocTM XRS+ system (Bio-Rad, Hercules, CA, USA).

## Statistical analysis

Statistical analyses were performed using GraphPad Prism 7.0 software (GraphPad, Inc., La Jolla, CA, USA) and using Student's *t*-test as indicated in the individual figure legends. *P* values <0.05 were considered significant. Error bars represent the standard error of the mean as noted in the individual figure legends.

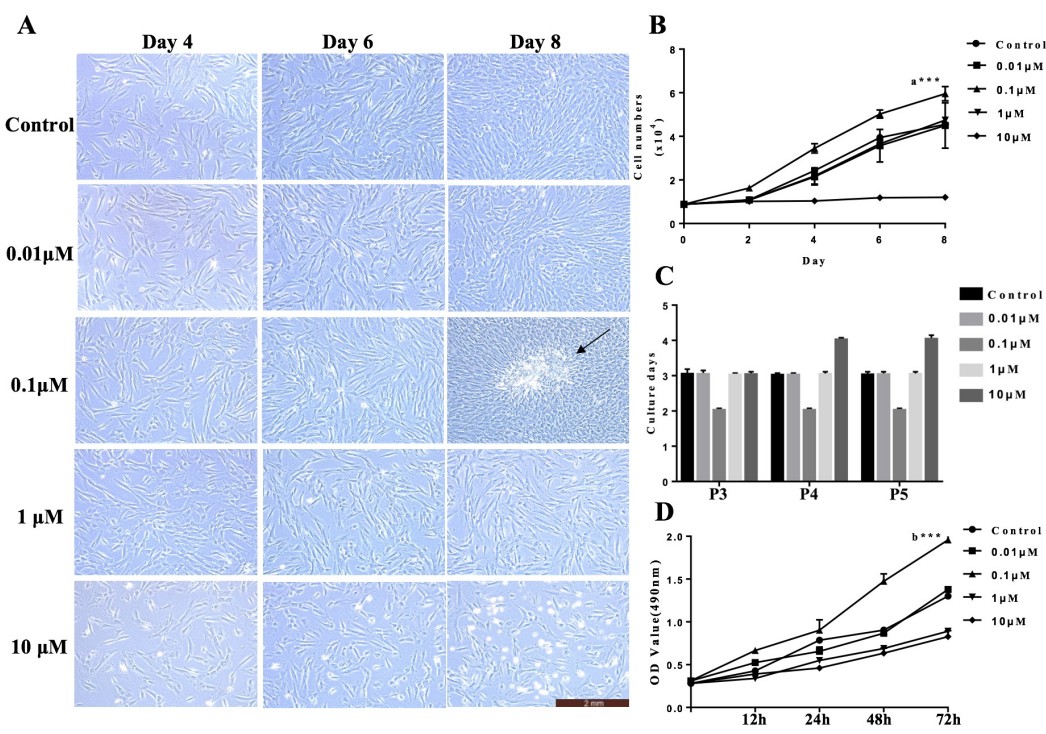

**Figure 1  VB-1 facilitates the proliferation of human DPCs.** (A) Morphology of human DPCs treated with VB-1 (0–10 μM) at indicated days. Arrow indicates colony growth of DPCs. (B) Human DPCs (1 × 10$^4$ cells) were plated in 24-well dishes and cultured in the presence of different concentrations of VB-1 (0–10 μM) for 8 days. Growth curves indicate the mean of three independent experiments (±SEM). (C) Culture days per passage of human DPCs treated with VB-1 (0–10 μM). Experiments were carried out in triplicates. (D) OD value of human DPCs (4 × 10$^3$ cells) were plated in 96-well dishes and cultured in the presence of different concentrations of VB-1 (0–10 μM) for 3 days. Data are reported as mean+SEM. Student's $t$-test was used to compare data. *$P < 0.05$, **$P < 0.01$.

## RESULTS

### VB-1 promotes the proliferation of human dermal papilla cells (hDPCs)

To assess the effects of VB-1 on cultured hDPCs, we first examined the proliferation of cells treated with different doses of VB-1 (0, 0.01, 0.1, 1, and 10 μM). During the 8 days of culture, the number of expanded hDPCs was greater in the 0.1 μM VB-1 group than in the control group (0 μM), and there was no significant difference observed at 0.01 and 1 μM. However, a high concentration of VB-1 (10 μM) dramatically suppressed the proliferation of hDPCs, and may have increased the apoptosis of cells (Figs. 1A and 1B). And these results were further confirmed by MTS assay in hDPCs treated with different doses of VB-1 (Fig. 1D). Interestingly, we also found that 0.1 μM VB-1 contributed to colony growth (Fig. 1A), indicating an increase in the hair-inducing capacity of hDPCs (*Osada et al., 2007*). Similarly, the number of culture days per passage for hDPCs were lower in medium supplemented with 0.1 μM VB-1 compared to in other groups (Fig. 1C). Collectively, these

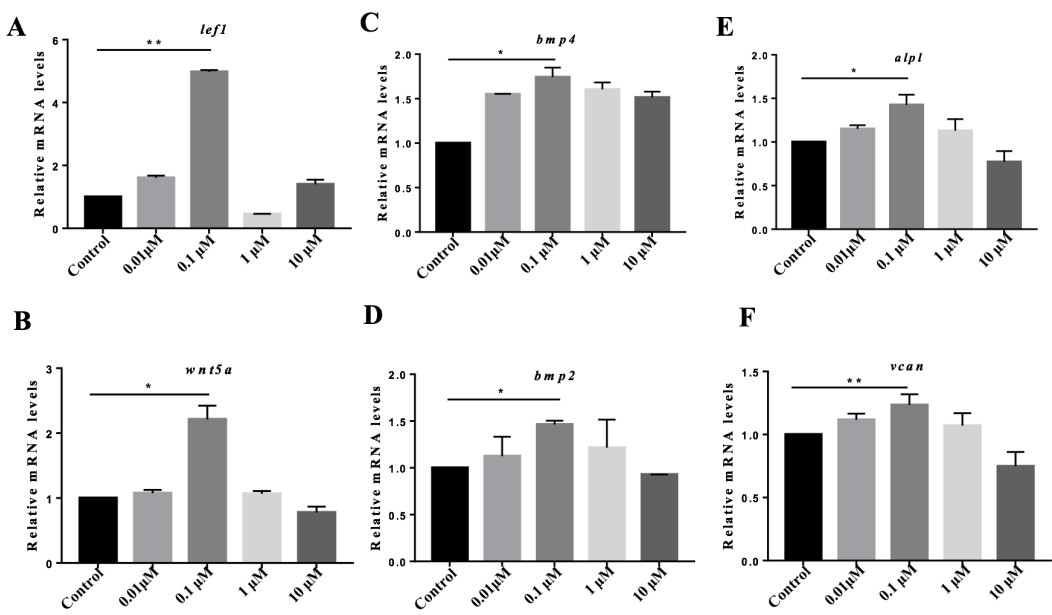

**Figure 2** **VB-1 increases the expression of the signature genes of human DPCs.** (A–F) Dose-dependent effects (0–10 μM) of VB-1 on *lef1*, *wnt5a*, *bmp2*, *bmp4*, *alpl* and *vcan* mRNA expression, in human DPCs cultured for 24 h. Data are shown as the ratio of the respective gene expression to gapdh mRNA expression. Experiments were carried out in triplicates. Data are reported as mean+SEM. Student's *t*-test was used to compare data. *$P < 0.05$, **$P < 0.01$.

results suggest that VB-1 facilitates the proliferation of hDPCs and improves hair-inducing abilities of these cells.

## VB-1 improves hair-inducing properties of hDPCs

Previous studies have suggested the indispensable roles of DPCs in hair follicle reconstruction assays *in vitro*. However, DPCs lose their hair-inducing properties quickly during culture (*Zhang et al., 2012*; *Zhang et al., 2015*; *Yang & Cotsarelis, 2010*; *Ohyama et al., 2010*), greatly limiting their applications for hair reconstitution. To determine whether VB-1 affects the hair-inducing ability of hDPCs, we treated cultured hDPCs with different concentrations of VB-1 (0, 0.01, 0.1, 1, and 10 μM). Our results showed that the Wnt signaling-associated signature genes of DP, *lef1*, and *wnt5a*, *alpl* and *vcan* were clearly upregulated in hDPCs treated with 0.1 μM VB-1 compared to in other groups (Figs. 2A, 2B, 2E and 2F). VB-1 also increased the expression of *bmp2* and *bmp4* (Figs. 2C and 2D), two additional markers (*Ohyama et al., 2012*). These findings indicate that VB-1 can promote the hair-inducing ability of hDPCs.

## VB-1 activates Wnt/β-catenin signaling in hDPCs

To define the roles of VB-1 in hair growth, we investigated its effects on the expression of hair growth-related genes in hDPCs. We found that compared to the control group, *dkk1* was significantly down-regulated by 0.1 μM VB-1 treatment and upregulated in the presence of 10 μM VB-1 (Fig. 3A). However there was no statistically significant difference in the expression of *tgf-β1*, another hair growth-related gene (Fig. 3B). By immunoblot analysis,

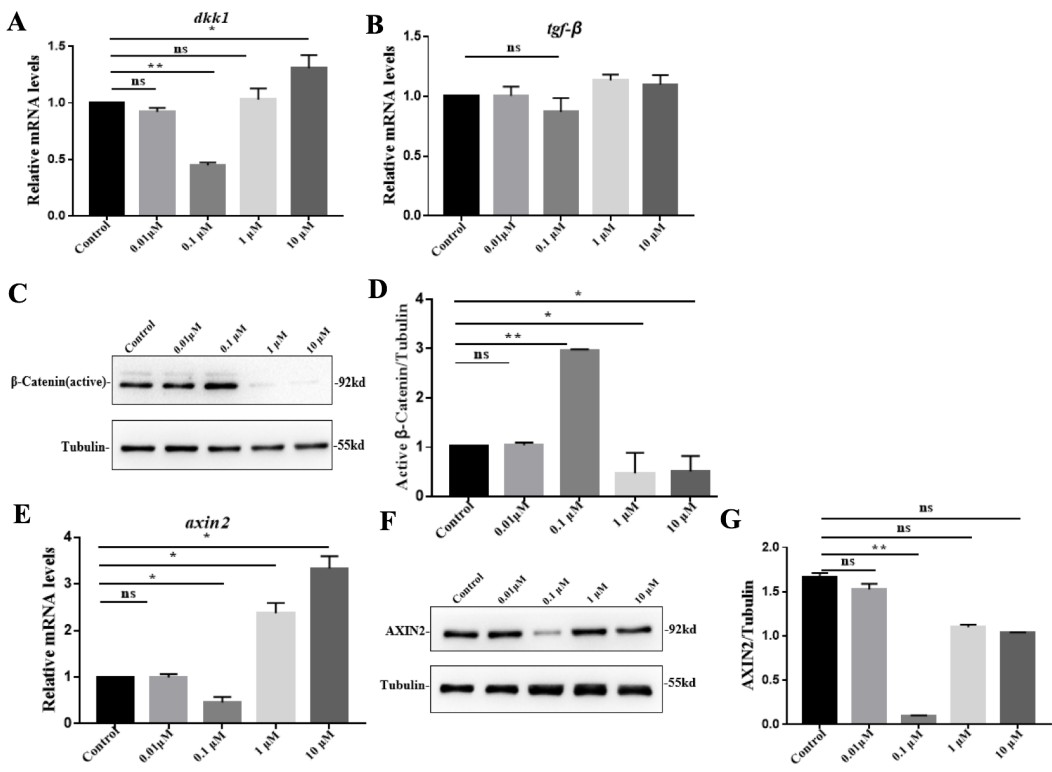

**Figure 3** **VB-1 promotes Wnt/β-catenin signaling in human DPCs.** (A) Concentration-dependent effects (0–10 μM) of VB-1 on *dkk1* mRNA expression in human DPCs cultured for 24 h. (B) Concentration-dependent effects (0–10 μM) of VB-1 on *tgf-β* mRNA expression in human DPCs cultured for 24 h. (C) Immunoblotting analysis of active β-Catenin expression in hDPCs treated with VB-1 (0–10 μM) for 24 h. (D) Quantification of active *β*-Catenin protein expression. (E) Real-time PCR analysis for gene expression of *axin2* in hDPCs treated with VB-1 (0–10 μM) for 24 h. (F) Immunoblotting analysis of AXIN2 expression in hDPCs treated with VB-1 (0–10 μM) for 24 h. (G) Quantification of AXIN2 protein expression. Experiments were carried out in triplicates. The typical blot was presented and quantification of three independent experiments is shown for C and F. Data are reported as mean+SEM. Student's $t$-test was used to compare data. *$P < 0.05$, **$P < 0.01$, "ns" indicates no significant difference.

we showed that the active *β*-Catenin protein level increased in hDPCs treated with 0.1 μM VB-1, while it decreased in hDPCs exposed to a high concentration of VB-1 (Fig. 3C). Moreover, the mRNA level of *axin2*, a negative regulator of Wnt/β-catenin signaling (*Fancy et al., 2011*), was reduced in hDPCs treated with 0.1 μM VB-1; high concentrations of VB-1 (1 and 10 μM) increased the expression of *axin2* (Fig. 3E). Suppression of AXIN2 by VB-1 in a dose-dependent manner was confirmed by immunoblotting analysis (Figs. 3F and 3G). These results suggest that VB-1 induces Wnt/β-catenin signaling activation in hDPCs in a concentration-dependent within a certain concentration range manner.

## VB-1 promotes hair shaft elongation in cultured human hair follicles

Because DP is essential for the regulation of hair growth, we further explored the possible effects of VB-1 on hair shaft elongation. Human scalp hair follicles were isolated and

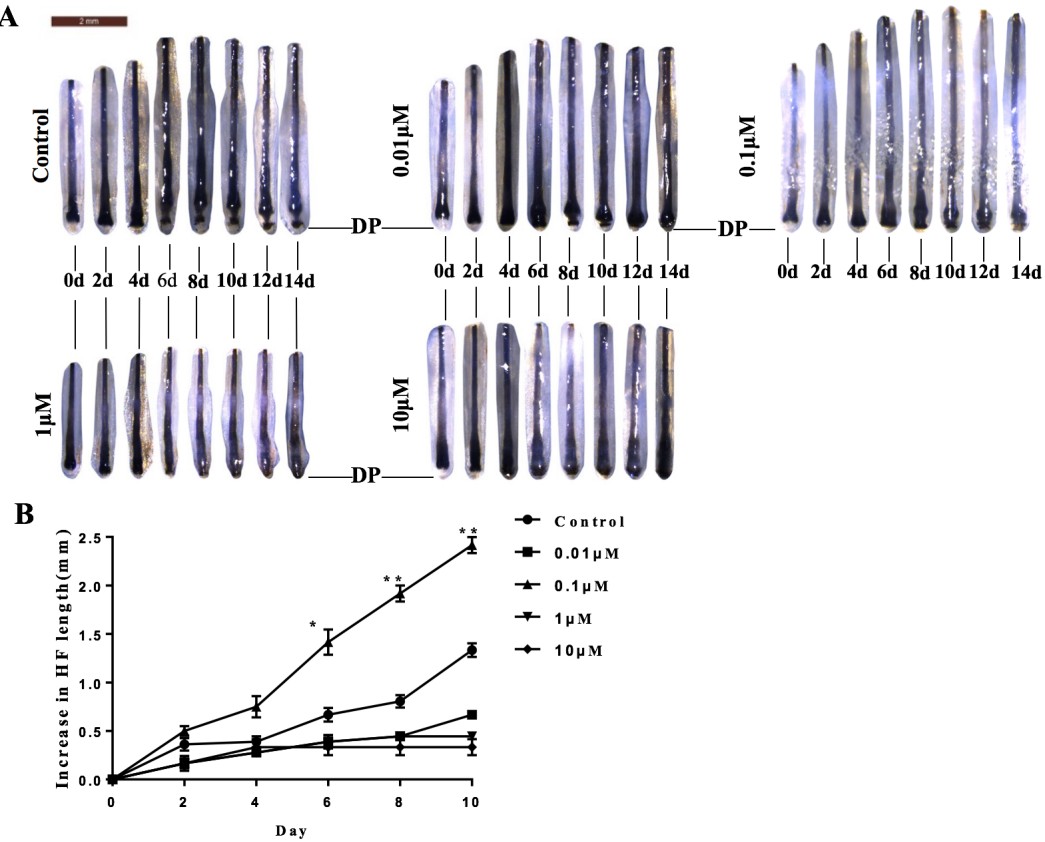

**Figure 4 VB-1 promotes the elongation of hair shafts in cultured hair follicles.** Isolated human scalp hair follicles were cultured for 14 days in the presence of different doses of VB-1. (A) Typical pictures of the hairs at day 0, 2, 4, 6, 8, 10, 12, 14. (B) Data are presented as the elongated length of the hair follicles treated with VB-1. Data are reported as mean +SEM. Student's $t$-test was used to compare data. *$P < 0.05$, **$P < 0.01$.

cultured in the absence or presence of VB-1. We found that 0.1 μM VB-1 significantly facilitated the elongation of hair shafts in cultured human hair follicles (Figs. 4A and 4B).

## DISCUSSION

The number of individuals currently suffering from hair thinning or balding, such as androgenetic alopecia, is increasing. Although numerous products claim to be useful for treating hair loss, they have sexual-related side-effects and unpredictable efficacy (*Varothai & Bergfeld, 2014*; *Rousso & Kim, 2014*; *Rogers & Avram, 2008*). Therefore, it is extremely important to develop new therapies for treating hair loss. In this study, we showed that VB-1 promotes the proliferation of hDPCs and partially restores hair-inducing properties. In HF organ culture, we demonstrated that VB-1 facilitates hair shaft elongation in cultured human scalp hair follicles, which may have resulted from the activation of Wnt/β-catenin signaling in hDPCs.

It has been shown that lignans exert anti-cancer activities by arresting cancer cells in the G2/M phase of the cell cycle and subsequently inducing apoptosis (*Zhou et al., 2009*; *Xin et al., 2013*). Interestingly, our data showed that VB-1 can promote the proliferation of hDPCs at a low dose; however, high dose of VB-1 inhibits the growth of these cells, which may be a result of the arrest of the cell cycle induced by high concentration of VB-1 as previously described in the cancer cells (*Zhou et al., 2009*; *Xin et al., 2013*), and the underlying molecular cues may need future study.

Adult human hair follicle reconstruction has become an attractive strategy for regenerative medicine, in which the roles of DP in epithelial-mesenchymal interactions that induce hair follicle neogenesis are indispensable (*Morgan, 2014*; *Higgins et al., 2013*). However, DPCs lose their hair-inducing properties quickly during culture *in vitro*, limiting their applications for hair follicle reconstitution (*Ohyama et al., 2012*). In the present study, we demonstrated that VB-1 increases the expression of human DP signature genes, such as *lef1*, *wnt5a*, and *bmp2*. Further studies are required to focus on whether VB-1 is suitable for long-term culture of hDPCs while maintaining their hair-inducing abilities.

Wnt/β-catenin signaling has been shown to be essential for hair morphogenesis and cycling (*Lien et al., 2014*; *Plikus & Chuong, 2014*), and its activation in the dermal papilla contributes to the proliferation and differentiation of hair follicle stem cells, thus initiating the anagen phase of the hair cycle (*Morgan, 2014*; *Enshell-Seijffers et al., 2010*; *Li, Jiang & Chuong, 2013*). Our data showed that VB-1 significantly upregulated Wnt/β-catenin signaling in hDPCs in a certain dose-dependent manner range. These observations suggest that VB-1 promotes hair growth by modulating Wnt/β-catenin signaling in hDPCs. Responding to active androgen receptor signaling, hDPCs produce a variety of paracrine factors such as *dkk1* and *tgf β-1*, impairing the proliferation and differentiation of hair follicle stem/progenitor cells, thus resulting in progressive HF miniaturization, a major characteristic of androgenetic alopecia (*Kwack et al., 2008*; *Inui & Itami, 2011*; *Shin et al., 2013*; *Ceruti, Leirós & Balañá, 2017*). Our results demonstrate that VB-1 decreases the expression of *dkk1* in cultured hDPCs.

The results of the present study show that VB-1 promotes hair shaft elongation in cultured human hair follicles in a concentration-dependent manner within a certain concentration range. Thus, VB-1 may be an effective therapy for the treatment of alopecia. However, further basic and clinical studies are required to verify the results presented in this study, and more practical dosing of VB-1 in the management of hair loss must be determined.

## CONCLUSIONS

Our findings strongly suggest that VB-1 augments Wnt/β-catenin signaling in human dermal papilla cells and significantly promotes the proliferation of hDPCs. Furthermore, VB-1 showed hair growth-promoting effects, indicating its potential as a new therapy for alopecia treatment.

## ACKNOWLEDGEMENTS

We thank Prof. Lunquan Sun (Center for Molecular Medicine, Xiangya Hospital, Central South University, China) and colleagues for their generous support throughout this work.

### Funding

This work was supported by the National Natural Science Foundation of China (grants 81602789, 81773351, 81472904, 81673086, 81573314). The funders had no role in study design, data collection and analysis, decision to publish, or preparation of the manuscript.

### Grant Disclosures

The following grant information was disclosed by the authors:
National Natural Science Foundation of China: 81602789, 81773351, 81472904, 81673086, 81573314.

### Competing Interests

The authors declare there are no competing interests.

### Author Contributions

- Jieshu Luo conceived and designed the experiments, performed the experiments, analyzed the data, contributed reagents/materials/analysis tools, prepared figures and/or tables, authored or reviewed drafts of the paper, approved the final draft.
- Mengting Chen and Yingzi Liu performed the experiments, contributed reagents/materials/analysis tools, prepared figures and/or tables, authored or reviewed drafts of the paper, approved the final draft.
- Hongfu Xie, Jian Yuan, Yingjun Zhou and Jinsong Ding contributed reagents/materials/analysis tools, prepared figures and/or tables, authored or reviewed drafts of the paper, approved the final draft.
- Zhili Deng and Ji Li conceived and designed the experiments, analyzed the data, contributed reagents/materials/analysis tools, prepared figures and/or tables, authored or reviewed drafts of the paper, approved the final draft.

### Human Ethics

The following information was supplied relating to ethical approvals (i.e., approving body and any reference numbers):

The Ethics committee of Xiangya Hospital, Central South University granted Ethical approval to carry out the study within its facilities (Ethical Application No: 201611609).

### Data Availability

The raw data are provided as Supplemental Files.

## Supplemental Information

Supplemental information for this article can be found online at http://dx.doi.org/10.7717/peerj.4737#supplemental-information.

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
