# Peer review of "Nature-derived lignan compound VB-1 exerts hair growth-promoting effects by augmenting Wnt/β-catenin signaling in human dermal papilla cells"

_PeerJ, doi:10.7717/peerj.4737_

## Round 0.1 · original submission · Minor Revisions

Dear authors,

After careful review, this manuscript need minor revisions. Please respond to the comments from the reviewers

Bingjin Li

·

Basic reporting

Writing. In general, the writing is good but can be improved in five parts.
1. Grammar errors. For example, the first sentence of the background in the abstract, the first sentence of the introduction, and the others in the text.
2. Some typos, for example, Abstract Method part, human dermal papilloma cells (hDPCs) is not correct.
3. Inconsistent abbreviations.
HFSCs in line 48, but to HFS in line 50.
DP cells, DPCs, human DPCs, and hDPCs.
V. negundo and Vitex negundo.
4. When you write RNA, please use lower case and italic. For example, when you check the mRNA levels of some genes like Lef1, you should write it in lower case and in italic. When you check its protein level, you should write it in capital.
5. Do not repeat your sentence in the abstract and introduction. For example, “Vitexins are a mixture of lignan compounds found in Vitex negundo, a widely used herb in China.”

Literature. More literature can be cited. For example, when you talk about hair reconstitution assay, the classical references should be cited. Also, references on Wnt signaling and its effect on proliferation of hair follicle should be cited.

Background/context. Can you briefly state why you would like to study the effect of vitexin compound 1 (VB-1) on hair follicle in the introduction part?

Arrangement of the manuscript. The structure of the manuscript is excellent, ranging from the phenotypic effect of VB-1 on in vitro cultured hDPCs, to the possible molecular mechanism, then to the in vitro organ culture effect. The figures are well arranged. However, the font in the figures can be enlarged to visualize.
Figure 3B should be Figure 3E; Accordingly, Figure 3E should be Figure 3B.

Results. The results are relevant to the hypotheses, although future study should be performed. For example, proliferation can be examined by immunostaining of some markers for Figure. Importantly, the VB-1-treated DP cells can be transplanted into the nude mice with epithelial cells, to observe if their hair induction ability is enhanced.

Experimental design

In the manuscript titled “Nature-derived lignan compound VB-1 exerts hair growth-promoting effects by augmenting Wnt/β-catenin signaling in human dermal papilla cells”, Luo et al., reported that vitexin compound 1 (VB-1) has a hair growth-promoting effect. This is an original primary research within the aims and scope of this journal.

The authors ask how VB-1 promotes hair growth. They first treated the cultured hDPCs with different doses of VB-1, and found that 0.1uM VB-1 promotes hDPCs proliferation. Then they identified that the expression of some DP signature genes and Wnt signaling pathway genes is changed in hDPCs after VB-1 treatment. They also treated the human scalp hair follicles with VB-1 in an in vitro organ culture system, and found that VB-1 significantly promotes hair follicle growth. This is a rigorous investigation using appropriate methods to examine the effect of VB-1 on hDPCs proliferation and on hair follicle growth.

Validity of the findings

The effect of VB-1 is more studied on other systems such as cancer cells, but not on hair follicle. So, this study has its novelty and impact on the potential future clinical application. The results and conclusions are appropriately made based on their scientific findings, and through enough replicate. However, there are still some statements needed to be improved.

1. Line 51-53, please make sure if the statement is correct. Are the outer root sheath and matrix cells generated from bulge stem cells with anagen initiation?

2. Result part. Figure 3, please clarify why the Axin2 mRNA levels are different from the protein levels after VB-1 treatment.
Also in Figure 3, the sampling times are inconsistent.
A descriptive error in the legend for figure 4B. This result does not show the percentage of hair follicle elongation.

3. In method part, authors wrote “150 anagen hair follicles isolated from 3 different volunteers were cultured with each concentration of VB-1”. However, in Figure 4 legend, authors wrote that “three independent experiments are shown (n=30)”. Which is the correct one?

4. Discussion part, line 205-206, authors wrote that “However, VB-1 at some low doses had proliferation-promoting effects in cultured hDPCs, suggesting that VB-1 represses the proliferation of cancer cells.” How and why can you make this suggestion which looks impertinent?

Additional comments

None

Reviewer 2 ·

Basic reporting

no comment

Experimental design

no comment

Validity of the findings

no comment

Additional comments

In this study, Luo and colleagues revealed that VB-1, extracted from Vitex negundo, a widely used herb in china, can significantly promotes the proliferation of human dermal papilla cells (hDPCs) in a concentration-dependent manner. They found that DKK1 is obviously down-regulated by VB-1 in hDPCs, and the increased active β-catenin and decreased Axin2 protein levels suggest VB-1 facilitates Wnt/β-catenin signaling in hDPCs in vitro. Moreover, the expression of DP signature genes are also upregulated after VB-1 treatment. They further showed that VB-1 promotes human HF growth by HF organ culture.

Overall the findings presented in this study are potentially interesting and novel. Before I can recommend publication, several issues require clarifications and/or further experimentations.


1. It will be informative if the authors can detect whether VB-1 affects the other hair growth-related genes (eg. TGFβ1).

2. The authors claimed that VB-1 improves the hair-inducing properties of human dermal papilla cells. It would be useful to assess whether VB-1 also promotes the expression of other important DP signature genes (such as ALPL and VCAN ).

3. The labels of figures (such as Fig3a, c, d, and f ) is indistinct. And the font in the figures should be uniform.

4. In Fig.1A and Fig.4A, please use “μΜ” instead of “uM”.

5. There are some typos and grammatical errors throughout the manuscript. It is essential to correct all the errors in the revised version of the manuscript.

Annotated reviews are not available for download in order to protect the identity of reviewers who chose to remain anonymous.

·

Basic reporting

1.The introduction should be written in more detail. Previous studies show that VB-1 induces apoptosis of cancer cells. This function of VB-1 should be mentioned in the introduction. In the last paragraph of introduction (lines 82-89), please clearly state your hypothesis and propose how to explore it.
2.The format of the reference does not meet the requirement of the journal. Please correct.
3.The time points shown in Figure 1A and 1B are inconsistent. Authors count the number of hDPCs at day 3, day 6 and day 9 in Figure 1A but at day 2, day 4 and day 8 in Figure 1B. Also, “After culture for 8 days” is more accurate than that “During the 9 days culture” in lines 152-154.
4.In Figure 2, the gene name should be written in italic.
5.In Figure 4A, the length of HFs treated with 10 μM VB-1 is shorter than that treated with 1 μM VB-1. But opposite results are showed in Figure 4B. Please clarify.
6.Please describe your statement more accurately in the manuscript. For example, “activity” can be changed to “capacity” in line 47.
7.Please write your abbreviations consistently and correctly. For example, in line 50, HFC should be HFSCs. In line 56, DPCs should be DP cells, or dermal papilla cells (DPCs). In line 67, WNT should be Wnt. In Figure 1A and Figure 4A, uM should be μM. Please check the whole manuscript and correct them.
8.Please discuss more why 0.1 μM VB-1 promotes hair growth and hDPCs proliferation but has an opposite effect in high concentration.

Experimental design

1.In the MM section, please briefly introduce how you did the real-time PCR. Please show the information for Tubulin antibody.
2.Authors aim to detect the effect of VB-1 on the proliferation of hDPCs by counting the number of hDPCs. I think the evidence is weak. Please add other methods to support this. For example, Brdu can be used to mark the proliferating cells.

Validity of the findings

Speculation is welcome, but should be more accurate. The authors aim to demonstrate that VB-1 promotes the proliferation of human dermal papilla cells (hDPCs) in a concentration-dependent manner (lines 149-150). Actually, VB-1 might promote the proliferation of hDPCs but not in a concentration-dependent manner because a high concentration of VB-1 inhibits the growth of hDPCs.

Additional comments

In this manuscript, authors showed that Vitexins promote hair growth by augmenting Wnt/β-catenin signaling in human dermal papilla cells. Although the manuscript needs to be improved, it is convincing that VB-1 can promote the growth of human hair follicles, which may provide a therapeutic strategy for treatment of alopecia.

---

## Round 0.2 · accepted · Accept

Dear Jieshu and Ji,

Thank you for your submission to PeerJ. Your manuscript - Nature-derived lignan compound VB-1 exerts hair growth-promoting effects by augmenting Wnt/β-catenin signaling in human dermal papilla cells - has been Accepted for publication. Congratulations!

#